# Harnessing Nanomedicine to Potentiate the Chemo-Immunotherapeutic Effects of Doxorubicin and Alendronate Co-Encapsulated in Pegylated Liposomes

**DOI:** 10.3390/pharmaceutics15112606

**Published:** 2023-11-09

**Authors:** Alberto Gabizon, Hilary Shmeeda, Benjamin Draper, Ana Parente-Pereira, John Maher, Amaia Carrascal-Miniño, Rafael T. M. de Rosales, Ninh M. La-Beck

**Affiliations:** 1Nano-Oncology Research Center, Oncology Institute, Shaare Zedek Medical Center, Jerusalem 9103102, Israel; hilary@szmc.org.il; 2Faculty of Medicine, The Hebrew University of Jerusalem, Jerusalem 9112102, Israel; 3King’s College London, School of Cancer and Pharmaceutical Sciences, Guy’s Cancer Centre, Great Maze Pond, London SE1 9RT, UK; benjamin.draper@ucl.ac.uk (B.D.); anacatpp@gmail.com (A.P.-P.); john.maher@kcl.ac.uk (J.M.); 4King’s College London, School of Biomedical Engineering & Imaging Sciences, St. Thomas’ Hospital, London SE1 7EH, UK; amaia.minino@kcl.ac.uk (A.C.-M.); rafael.torres@kcl.ac.uk (R.T.M.d.R.); 5Department of Immunotherapeutics and Biotechnology, Jerry H. Hodge School of Pharmacy, Texas Tech University Health Sciences Center, Abilene, TX 79601, USA; irene.la-beck@ttuhsc.edu

**Keywords:** liposome, doxorubicin, alendronate, co-encapsulation, chemotherapy, immunotherapy

## Abstract

Encapsulation of Doxorubicin (Dox), a potent cytotoxic agent and immunogenic cell death inducer, in pegylated (Stealth) liposomes, is well known to have major pharmacologic advantages over treatment with free Dox. Reformulation of alendronate (Ald), a potent amino-bisphosphonate, by encapsulation in pegylated liposomes, results in significant immune modulatory effects through interaction with tumor-associated macrophages and activation of a subset of gamma-delta T lymphocytes. We present here recent findings of our research work with a formulation of Dox and Ald co-encapsulated in pegylated liposomes (PLAD) and discuss its pharmacological properties vis-à-vis free Dox and the current clinical formulation of pegylated liposomal Dox. PLAD is a robust formulation with high and reproducible remote loading of Dox and high stability in plasma. Results of biodistribution studies, imaging with radionuclide-labeled liposomes, and therapeutic studies as a single agent and in combination with immune checkpoint inhibitors or gamma-delta T lymphocytes suggest that PLAD is a unique product with distinct tumor microenvironmental interactions and distinct pharmacologic properties when compared with free Dox and the clinical formulation of pegylated liposomal Dox. These results underscore the potential added value of PLAD for chemo-immunotherapy of cancer and the relevance of the co-encapsulation approach in nanomedicine.

## 1. Introduction

Co-encapsulation of multiple drugs in the same nanocarrier is a unique tool of nanomedicine offering multiple pharmacologic advantages, such as co-delivery in space and time of two or more agents, maximizing their additive or synergistic effects in cancer therapy or other fields of medical therapy. Formulating nanoparticles containing co-encapsulated drugs is an attractive strategy for the co-delivery of drugs with different mechanisms of action and non-overlapping toxicities [1]. Dual or multi-drug liposomes have been initially proposed by the group of Tolcher and Mayer [2], whose approach is based on screening in vitro for drug ratios that result in synergistic cytotoxicity. The field of co-encapsulation or co-delivery of multiple drugs in nanoparticles has attracted increased attention in recent years, particularly in cancer applications [2,3,4,5,6,7]. This approach includes examples of combinations of two cytotoxic drugs [8], one of which may be a prodrug [9], or one cytotoxic drug and an immunomodulatory drug [10]. The success of this approach in humans is exemplified by the FDA approval of CPX-351 (Vyxeos^®^), an optimized ratio of daunorubicin and cytarabine co-encapsulated in liposomes that significantly improved survival in acute myeloid leukemia patients (AML) when tested against the conventional treatment with the same drug combination in free form [11].

We have developed a pegylated liposome formulation with 2 active ingredients, alendronate (Ald) and doxorubicin (Dox), referred to as PLAD, that display very different mechanisms of action and have no overlapping toxicity [12]. The choice of Dox is well supported by a plethora of preclinical and clinical data asserting its compatibility with liposome formulations and its clinical value as an anticancer agent [13]. The choice of Ald is based on the multifaceted properties of aminobisphosphonates, including direct (reduced tumor cell invasion and proliferation) and indirect (reduced osteoclastic activity and angiogenesis) antitumor effects [14,15], along with immunological effects (increased activity of gamma-delta T cells and suppression of tumor-enhancing macrophages when formulated in liposomes) [16,17]. The lipid backbone of this formulation is very similar to the clinically approved formulation of pegylated liposomal Dox (PLD), known commercially as Doxil/Caelyx^®^ or LipoDox for the generic version. We have previously reported that PLAD significantly affects the composition and functionality of tumor-infiltrating immune cells [18]. We report here on various improvements in the pharmaceutical technology and characterization of the PLAD (Pegylated Liposomal Ald-Dox) formulation [12] and explore further aspects of its biological performance in vitro and in vivo.

## 2. Materials and Methods

### 2.1. Chemicals Sources

Hydrogenated soybean phosphatidylcholine (HSPC), Lipoid GmbH, Ludwigshafen, Germany; methoxy-polyethylene glycol-distearoyl-phosphatidylethanolamine (mPEG_2000_-DSPE), Bio-Lab Ltd., Jerusalem, Israel; cholesterol (Chol) and ammonium hydroxide, both from Sigma-Aldrich Israel Ltd., Rehovot, Israel; Alendronic acid (Ald), Tokyo Chemical Industry Co Ltd., Japan; Doxorubicin HCl (Dox), Teva Pharmaceuticals, Tel Aviv, Israel; Pegylated liposomal doxorubicin (PLD), either as Doxil/Caelyx™ (Janssen Pharmaceuticals, Beerse, Belgium) or as Lipodox (Taro Pharmaceuticals, Haifa, Israel).

### 2.2. Formulation of PLAD

Most of the experiments presented here were conducted with 2 large batches of PLAD (0.5–1.5 L) prepared at Nextar Chempharma (Ness Ziona, Israel) following a process similar to that reported previously [12] utilizing a 250 mM ammonium alendronate gradient in the same way as the classical ammonium sulfate gradient of PLD, resulting in effective and stable loading of Dox in liposomes [11]. Vesicle size was measured using dynamic light scattering (DLS) on a Zetasizer (Malvern Panalytical Ltd., Worcestershire, UK). Zeta potential measurements were performed at 25 °C using a Zetamaster (Malvern Panalytical Ltd.). Liposome samples were imaged using cryo-transmission electron microscopy (cryo-TEM). Sample preparation and examination by cryo-TEM were carried out at the Hebrew University Center for Nanoscience and Nanotechnology (Jerusalem, Israel) on an FEI Tecnai 12 G2 TEM, operated at 120 kV. Further details of the formulation methodology are as previously reported [11]. In a few experiments, we used small batches of PLAD (~50 mL) prepared in our laboratory as described before [12].

A summarized description of the formulation of PLAD follows. The PLAD formulation is prepared by the standard method of ethanol injection into an aqueous buffer containing a salt of ammonium alendronate (passive encapsulation), followed by extrusion, buffer exchange, and remote gradient loading of Dox based on a previously described method [12]. The lipid components: HSPC, mPEG-DSPE, and Chol at 55%, 40%, and 5% molar ratios, respectively, are dissolved in warm (60 °C) ethanol. This ethanol lipid solution is then mixed with an aqueous buffer of 250 mM ammonium alendronate salt, prepared by mixing a solution of 250 mM alendronic acid with ammonium hydroxide (25%) with a pH in the range of 6.2–6.8. After mixing and shaking for 1 h at 60 °C, the multilamellar vesicles obtained are downsized by serial extrusion in a high-pressure extruder (Lipex Biomembranes, Evonik Canada Inc., Burnaby, BC, Canada) at 60 °C through double-stacked polycarbonate 0.08 µm pore size membrane filters. Nonencapsulated Ald and residual ethanol are removed by tangential flow filtration (TFF) against a dextrose/Hepes buffer (5% dextrose with 17 mM sodium HEPES, pH 7.0). The liposomes are then remotely loaded with Dox with a gradient generated by ammonium alendronate (Figure 1A) by mixing with a solution of 10 mg/mL Doxorubicin HCl in dextrose/Hepes buffer and incubating for 30 min at 60 °C. Non-encapsulated Dox is removed by TFF. The liposome suspension is clarified by filtration through 0.45/0.22 µm-pore cellulose membranes. The doxorubicin concentration is then measured, and its final concentration in the formulation is adjusted to 1.0 mg/mL by further dilution with dextrose/Hepes buffer, after which the liposome product undergoes final sterilization by filtration through 0.22 µm-pore cellulose membranes.

### 2.3. Stability Assays

Formulation stability was assessed based on the exposure to either pooled expired human plasma as previously described [11] or to human serum albumin (HSA) from commercial sources. We have recently introduced this test based on the exposure of liposomes to albumin due to the need for a standardizable stability assay for the release of liposome batches for clinical use. The HSA-based liposome stability test is preferable to the plasma stability test. HSA is regulated as a biological pharmaceutical product and can be obtained in an aqueous solution, lyophilized, or more recently, as a recombinant product (Leveraging GMP-Grade Human Serum Albumin for Pharmaceutical Manufacturing (https://www.pharmasalmanac.com/articles/leveraging-gmp-grade-human-serum-albumin-for-pharmaceutical-manufacturing (accessed on 24 May 2019)). HSA lots are well characterized and have a uniform composition from lot to lot with a minimal content of impurities. The stability assay itself is similar to the former assay in plasma. We use a concentration range of 5% to 20% HSA, which, upon dilution after mixing with liposomes, results in a final concentration of 4% to 16% HSA. This range covers the physiologic concentration of albumin in plasma (4 g %) and above.

### 2.4. In Vitro Uptake and Cytotoxicity Assays

Uptake and cytotoxicity were tested on a variety of human and mouse carcinoma cell lines. Cells were plated and incubated with free or liposomal drugs for the measurement of uptake and cytotoxicity as described in prior references [12,19] and in the Results section.

### 2.5. Animal Studies

Female inbred BALB/c and outbred Sabra mice, 8–10 weeks old, were obtained from Harlan Biotech (Jerusalem, Israel). In vivo experiments were performed either at the Shaare Zedek Medical Center Animal Lab or at the Animal Facility of the Hebrew University-Giv’at Ram Science Campus. Animal experiments conducted in Israel were approved by the Animal Ethics Committee of the Hebrew University–Hadassah Medical School (research number: MD-20-15666-5). Animal experiments conducted in the UK were ethically reviewed and carried out in accordance with the Animals (Scientific Procedures) Act 1986 (ASPA) UK Home Office regulations governing animal experimentation with local approval from King’s College London Research Ethics Committee. For further details on animal studies and tumor models, see relevant sections of Results.

### 2.6. Determination of Dox in Pharmacokinetic and Biodistribution Studies

Mice were injected i.v. with an equal dose of free Dox, PLD, or PLAD based on doxorubicin content. For blood collection, mice were anesthetized by halothane or isoflurane inhalation, bled by eye enucleation (~1 mL blood per mouse), and immediately sacrificed by cervical dislocation. Blood was collected in heparinized tubes and centrifuged immediately to separate plasma from blood cells. Plasma levels of doxorubicin were measured fluorometrically after extraction from plasma with acidified isopropanol, as described previously [20]. Tissue biodistribution was assessed either in tumor-free or tumor-bearing mice. Tumors were generated by subcutaneous inoculation of tumor cell suspensions in the flanks or in the inter-scapular space.

### 2.7. Preparation of Radiolabelled PLAD ([^111^In]In-PLAD) and Formulation

[^111^In]In-PLAD was radiolabeled as previously reported in similar PEGylated liposomes [21] using an ionophore approach with [^111^In]In(oxinate)_3_ produced using the chloroform method [22]. The final formulation injected per mouse contained 50 µg radiolabeled PLAD liposomes and plain (drug-free) HSPC/CHOL/mPEG2000-DSPE liposomes (FormuMax, Stratech Scientific Ltd., Ely, UK) with similar lipid composition to PLAD to a total of 4 µmol of lipids in 130–150 µL of PBS.

### 2.8. SPECT/CT Imaging Study with Ex Vivo Biodistribution and Tumor Autoradiography

We conducted an imaging study on a WEHI-164 subcutaneous mouse tumor model. After 10–14 days of subculturing, WEHI-164 cells were harvested, and two million cells were inoculated subcutaneously unilaterally in the shoulder of BALB/c mice 8–9 weeks old (Charles Rivers, UK). On day 9, after inoculation, the mice were injected i.v. by tail vein bolus injection with ca. 10 MBq [^111^In]In-PLAD. Mice underwent SPECT/CT imaging at 30 min, 24 h, 48 h, and 72 h post-injection. SPECT imaging was performed with a four-headed multiplexing multipinhole NanoSPECT/CT (Mediso, Hungary) using Aperture 3 (1 mm pinholes). A 96 mm field of view comprising the animal from the tip of the nose to the end of the back legs was used, with an energy peak of 171 and 245 KeV ± 10% keV. The acquisition time was adjusted to range from 40 to 80 s, increasing it to compensate for the isotope decay at later time points. A 21-min 360-frame CT image was performed immediately before or after the SPECT acquisition. Reconstruction of the images was performed, including attenuation correction, using the software Scivis HiSPECT version 1.4.1876 (Invicro, Needham, MA, USA) with standard parameters. Reconstructed data from SPECT and CT were co-registered using ViVoquant 2020 (Invicro, Needham, MA, USA) for further analysis and interpretation.

For analysis of ex vivo biodistribution after completion of imaging, the tissues were collected, weighted, and counted in the 1282 CompuGamma gamma counter (LKB Wallac, Gungahlin, Australia), alongside standard samples of known radioactivity. For autoradiography studies, tumors were snap-frozen and cut into 45 μm slices for autoradiography. The tumor slices were set against imaging plates (GE, Chalfont St Giles, UK) for 3 days, and autoradiograms were obtained using an Amersham Typhoon 5 (GE, UK) analyzer system with a resolution of 25 μm and a sensitivity of PTM of 4000.

### 2.9. Toxicity Studies

These studies were done in tumor-free BALB/c mice receiving weekly i.v. injections of PLD or PLAD. Mice were observed, weighed weekly (×3), and followed for up to 60 days.

### 2.10. Antitumor Efficacy

BALB/c female mice (~8–10 weeks) were inoculated with M109R mouse tumor cells (10^6^ cells) or Wehi-164 mouse tumor cells (10^6^ cells) s.c. in the left or right flank. In the 4T1 model, tumor cells (10^5^ cells) were injected into the right hind footpad. When tumors became palpable, free drug or liposomal drug treatment was injected i.v. in the tail vein, while anti-PD1 mouse antibodies were injected i.p. according to the schedule of each specific experiment. Mice were monitored at least twice per week for body weight and for tumor size with precision calipers. Tumor growth was followed for up to 60 days. Statistical analysis was done using Prism software version 9 (Graphpad, San Diego, CA, USA).

### 2.11. Therapeutic Studies Combining PLAD with Gamma-Delta T Cell Transfer

These experiments were done as described previously for a human epithelial ovarian cancer model [23] treated with gamma delta (subset Vγ9 Vδ2) T cells, except that the tumor model used here was the MDA-MB-231, a triple-negative human breast cancer model, and in addition to PLA, PLAD was also tested. Gamma delta T cells were obtained from the blood cells of healthy donors, expanded in vitro, and collected for in vivo studies as described previously [23].

## 3. Results

### 3.1. Formulation and Characterization of PLAD

For details on the PLAD formulation process, see the Methods section above. The concentrations of the liposome components of PLAD for two successive batches are listed in Table 1. All values obtained fell within a pre-specified target range considered to be acceptable for batch release. The potency of the formulation is based on the Dox content of PLAD, which is measured by a previously described HPLC assay [12]. Ald concentration is based on a phosphorous assay of the upper phase of a Folch extraction of the liposomes, as described previously [11]. PLAD average vesicle size, as measured by dynamic light scattering, is 90–100 nm with narrow polydispersity (PDI < 0.15). CryoTEM photographs of the PLAD formulation reveal spherical vesicles with intravesicular packs of rods resulting from the crystallization of the Ald-Dox complexes. Unlike PLD, no oval-shaped liposomes are seen in PLAD, and the PLAD rods appear to be shorter and more loosely packed than the Dox-sulfate rods of PLD (Figure 1B).

Upon storage at 4–8 °C, PLAD is highly stable in aqueous buffer suspension, retaining >97% of Dox in encapsulated form with vesicle size remaining stable for >18 months. In the past, we have used Sepharose columns to separate released free drugs from liposomal drugs [12]. A more convenient and accurate method to follow up on the stability of encapsulation is centrifugation of a liposome sample using Vivaspin^®^ ultrafiltration tubes (Sartorius, UK) with the appropriate MW cutoff (300 Kd) such that only free drug passes through the filter and can be quantified by the relevant methods: phosphorus assay for Ald and fluorescence assay for Dox. Based on this assay and on DLS particle size analysis, we found no significant leakage of Dox or Ald and no significant change in vesicle size or polydispersity, suggesting that these critical parameters of the PLAD formulation are stable over the course of ~2 years (Data on file, Levco Pharmaceuticals, Jerusalem, Israel).

### 3.2. Stability of PLAD in Biological Fluids

The stability of the PLAD formulations was assessed in vitro with a plasma stability assay by exposure to human plasma for 2 h at 37 °C. This test gives a good prediction of the degree of stability in vivo in circulation, although it has serious limitations as a standard test for drug development because of the variability of plasma sources, which are usually obtained from expired batches of fresh frozen plasma before they are discarded from the blood bank.

We have shown in the past that no leakage of Ald occurs in plasma [12]. In fact, Ald is very hydrophilic and cannot cross a cholesterol-rich solid bilayer at 37 °C, as in the case of HSPC-containing PLAD liposomes. In addition, most of the Ald is complexed with Dox and precipitated in the liposome water phase. Release of Ald requires first dissociation from the doxorubicin complex and then a breakdown of the liposomal bilayer integrity. Unlike Ald, Dox may leak from liposomes if the proton gradient is lost because of its amphipathic nature, even if the liposomal bilayer remains intact. We, therefore, chose to examine the leakage of Dox as a surrogate marker of liposome stability in biological fluids.

Figure 2A shows the release of Dox in fractions of eluent collected from a Sepharose column after incubation of the liposomes in human plasma and in buffer. Nearly all the drugs remain in liposome-associated form and elute together with liposomes in fractions 4–6, while plasma proteins elute mostly in fractions 7–11. There was a minor and insignificant difference between the elution profiles in plasma and buffer. This indicates that drug leakage in plasma is minimal and probably insignificant.

As an alternative to plasma, we have used commercial sources of human serum albumin (HSA), which is well characterized and has a uniform composition from batch to batch. As seen in Figure 2B, incubation of PLAD in 4–16% HSA under the same conditions as plasma resulted in negligible leakage of Dox, indicating that PLAD is highly stable when exposed to a protein-rich fluid and maintains the gradient that holds the drug in the vesicle interior.

### 3.3. In Vitro Cell Studies with PLAD: Uptake and Cytotoxicity

Dox uptake studies in a variety of tumor cell lines indicate great variability in liposomal drug uptake, but, along with that, there was a trend toward higher uptake of PLAD when compared with PLD in all cell lines (Figure 3A). Since drug leakage is negligible under these conditions for both formulations, the uptake of Dox is probably related to the number of vesicles taken up by the cells. The Dox/phospholipid ratio is higher in PLD than in PLAD and therefore cannot explain this difference in drug uptake. It is tempting to speculate that these small differences may be related to other characteristics such as differences in vesicle shape, aspect ratio, or membrane rigidity between PLAD and PLD [24].

We also looked at liposome uptake when raising the temperature to 42 °C in KB cells, a cell line that has a high endocytic activity for liposomes. Interestingly, as seen in Figure 3B, liposomal drug uptake was greatly increased with both liposomal formulations (18 to 24-fold) as compared with free drug (~4-fold). PLD and PLAD are both high-Tm liposomes that are unlikely to leak drugs at temperatures below 50 °C. This has been confirmed in experiments with the grafting of ligands onto Dox-pre-loaded liposomes at 45 °C [25].

Therefore, the increased uptake of liposomal drugs with a moderate rise in temperature is probably related to an increase in endocytic activity. If these observations are confirmed in vivo, they may have translational relevance and support the use of liposomal drugs rather than free drugs with regional hyperthermia.

As expected, PLAD and PLD were much less cytotoxic than free Dox, which is always the case in vitro for stable liposome formulations [26]. The in vitro cytotoxicity of PLAD is consistently superior to that of PLD on several mouse and human carcinoma cell lines (Table 2, see also growth inhibitory curve in Appendix A), with a broad variation in sensitivity to Dox. This increased cytotoxicity may be the result of the slight increase in in vitro uptake of PLAD as compared with PLD (Figure 3). A simple additive effect of Ald is unlikely since free Ald and, more so, liposomal Ald have little or no in vitro cytotoxic effect in the pharmacological concentration range [12,27]. However, Ald may sensitize the cells to Dox once it becomes available in the intracellular compartment through the inhibition of the mevalonate pathway at the level of FPP synthase [28]. This will translate into the synergistic cytotoxicity of PLAD, as shown for another aminobisphosphonate encapsulated in nanoparticles [29].

### 3.4. Pharmacokinetics and Biodistribution

PLAD demonstrated a prolonged circulation time in mice, slightly lower than PLD (Figure 4A), with a difference of minimal significance based on total plasma Dox concentrations. The tissue Dox levels were increased moderately in the liver, markedly in the spleen, and somewhat decreased in the kidneys when PLAD was compared with PLD (Figure 4B). As previously observed in the Wehi-164 model [18], we found a non-significant increase in tumor drug content when PLAD is compared with PLD in the M109 tumor model. Both liposome formulations dramatically increased the amount of Dox measured in tumor tissue when compared with free Dox (Figure 4C). An additional observation in PLAD-injected mice, similar to what was reported with PLD [30], was a non-significant trend to lower drug levels per gram tumor as the tumor size increased (Figure 4D).

### 3.5. Imaging Studies of PLAD in Tumor-Bearing Mice

To further investigate the biodistribution of PLAD in tumor-bearing mice, we conducted an imaging study (SPECT-CT) in BALB/c mice inoculated with the WEHI-164 tumor model. As indicated in the Methods section, PLAD was radiolabeled with the gamma-emitter indium-111 (^111^In) using a previously published method [21] to form [^111^In]In-PLAD. Each mouse received a total dose of 4 µmol of lipids by combining [^111^In]In-PLAD with empty liposomes of the same composition and physicochemical properties (size and zeta potential) but lacking doxorubicin/alendronate. After intravenous injection, mice were imaged by SPECT-CT (Figure 5A) for up to 72 h, and, at the end of the study, an ex-vivo biodistribution of analysis in mice injected with [^111^In]In-PLAD was performed (Figure 5B).

The study revealed high and heterogeneous accumulation of PLAD in the tumor (Figure 5A), with an average uptake value of 40.4 ± 28.7% Injected Activity (IA)/g at 72 h and the highest tumor uptake value of 101% IA/g in a very small tumor. As expected from previous biodistribution data in mice, the spleen was the organ that had the highest uptake at this time point, followed by the tumor and liver (Figure 5B). Further analysis of the results indicates a higher uptake in tumors of smaller size in comparison with larger ones (Figure 5C), as suggested for liposomal drugs in Figure 4D and consistent with previously published observations on liposome biodistribution [31]. The intratumoral distribution was heterogeneous, with a higher concentration of radiolabeled PLAD at the edge of the tumor, as observed via autoradiography studies of small slices of tumor tissue (see inset in Figure 5A).

### 3.6. Toxicity Study

An experiment comparing the toxicity of PLD and two batches of PLAD in tumor-free outbred Sabra female, 7-week-old mice. Mice were injected i.v. with a dose of Dox close to the maximal tolerated dose of PLD, 10 mg/kg, in two successive weekly injections and followed for 6 more weeks. As seen in Figure 6, the weight curves of PLAD-injected mice rose shortly after treatment, unexpectedly suggesting fluid accumulation, and then dipped, but not more than 15%. One mouse out of 8 injected with PLAD died on day 14. Hair loss was also noted in one mouse injected with PLAD. All other mice survived, recovered, and gained weight normally. The weight gain of PLD-injected mice was transiently affected, but otherwise, there was no other sign of toxicity. Based on these results, PLAD seems to be slightly more toxic than PLD. We therefore chose to conduct therapeutic studies with dose levels ≤8 mg/kg in immunocompetent mice.

### 3.7. Therapeutic Activity of PLAD

Our former observations in the M109R and 4T1 tumor models in immunocompetent mice [12] were reproduced with the current optimized formulation of PLAD. PLAD was superior to PLD in these models. In the 4T1 tumor model, a significant number of complete tumor regressions or cures were achieved with PLAD (5/9) compared with PLD (1/9) (Figure 7A–C). In addition, we conducted therapeutic studies on the WEHI-164 mouse sarcoma model. In this highly Dox-sensitive model, PLAD and PLD demonstrated great efficacy with complete tumor regression in 100% and 90%, respectively, of mice inoculated with the WEHI-164 sarcoma model implanted subcutaneously in BALB/c mice. Free Dox treatment was also highly efficacious but resulted in fewer (70%) complete tumor regressions (Figure 7D–G).

We then explored the therapeutic activity of PLAD and PLD in combination with a mouse anti-PD1 antibody in the Dox-resistant M109R tumor model. As seen in Figure 8, treatment with PLAD and anti-PD1 resulted in the best outcome, with the smallest tumors observed at the end of the study. Free Dox is ineffective in this highly multidrug-resistant tumor model [32].

### 3.8. PLAD and Gamma-Delta T Cells

Our previously published work with PLA in combination with adoptively transferred gamma-delta (Vγ9 Vδ2) T lymphocytes from human donors in an in vivo human tumor model demonstrated a significant antitumor effect of this combination [23]. Moreover, PLA was shown to increase the number of infused gamma-delta T cells localizing in tumors [33]. Subsequent in vitro studies with human breast cancer and AML cell lines indicated that PLAD is a strong activator of gamma-delta T cells (US Patent #10,085,940 [34]).

We therefore conducted experiments to investigate the activity of PLAD in combination with human gamma-delta T cells in a human breast cancer mouse tumor model. As seen in Figure 9A,B, the best therapeutic outcome was seen with the combination of gamma-delta T cells and PLAD in combination with human gamma-delta T cells. PLAD alone was also highly active, but Vγ9 Vδ2 T lymphocytes as a single modality treatment or in combination with PLA were clearly less effective. No significant toxicity (weight loss, general appearance) was observed when PLAD was combined with gamma-delta T cells in the course of the experiment. For a detailed figure with all weekly measurements of bioluminescence, see Appendix A.

## 4. Discussion

Co-encapsulation in a stable nano-formulation of two active agents, preferably with non-overlapping toxicities and synergistic effects, is a unique advantage of nanomedicines. By space and time co-delivery of two drugs with otherwise different pharmacokinetic-biodistribution profiles, we can exploit combination therapy at its best and achieve optimal synergistic activity. As mentioned in the Introduction section, a clinical example is a liposome-based formulation of cytarabine and daunorubicin at an optimized 5:1 drug-to-drug ratio, known as Vyxeos™, approved for the treatment of adult AML [35]. In this formulation, the liposome carrier controls and nearly equalizes the pharmacokinetics of both drugs [36]. There are other examples of co-encapsulated drugs in liposomes and polymeric formulations with positive results in animal models [37,38,39]. However, in most instances, co-encapsulation consists of two cytotoxic drugs, in contrast to our PLAD formulation, in which two drugs with non-overlapping mechanisms of action and toxicity profiles are co-encapsulated. While this approach is pharmaceutically and regulatory-wise challenging, it is a unique advantage of nanomedicine and holds promise for future applications [30].

The starting point for the formulation is the well-known PLD formulation approved for ovarian and breast cancer and widely used in the clinic for more than 20 years with a great safety record [13]. Despite the important pharmacologic advantages of PLD, its impact and added value on the survival of cancer patients have been modest. A number of reasons have been invoked for this apparent discrepancy between the preclinical and clinical results [13]. We hypothesize that nanodrugs, particularly those that deliver immunogenic cell death inducers such as doxorubicin [40], are more suitable and effective than conventional chemotherapy in combination with immunotherapy given their affinity and their putative suppressive effect on tumor-associated macrophages (TAM), which tend to have an overwhelming M2 tumor-promoting effect [41,42]. Therefore, one way to improve the performance of PLD may be in combination with immunotherapy. Clinical studies with a combination of PLD and immune checkpoint inhibitors are still in an early phase, but their initial results are encouraging [43,44,45,46]. In parallel, we postulate that the combination of Dox with an immunomodulator and TAM-suppressor drug such as ALD in the same liposome, as done in PLAD, should improve the synergistic effect with anti-PD1 antibodies and perhaps other checkpoint inhibitors, as suggested by a recent study showing that PLAD has a stronger raising effect on the M1/M2 ratio when compared with PLD [18].

In the said study by Islam et al. [18], PLAD and PLD were found to shift the balance between various immune cell types and their functionality, creating changes in the TME conducive to an improved antitumor response [18]. These effects were absent in free Dox-treated mice. The effects of PLAD were generally stronger than those of PLD, particularly the association of PLAD with TAM, suppression of TAM activity, and relative increase in the ratio of M1 over M2 macrophages. Besides their effects on TAM, treatment with aminobisphosphonates, particularly in liposomal form, results in the formation of phospho-antigens that stimulate a natural immunity response against cancer mediated by Vg9Vd2 gamma-delta T cells. In primates, most circulating gamma-delta T cells express the Vg9Vd2 TCR, enabling their HLA-independent activation and expansion by nonpeptide phospho-antigens [23]. This provides a qualitative and unique advantage to Ald-containing liposomes and, by extrapolation, to PLAD over PLD. Furthermore, treatment with liposomal Ald significantly increased the homing of adoptively transferred gamma-delta T cells to tumors in a mouse model [33].

PLAD offers other advantages in the field of theranostics since Ald is a potent chelator of various metal radionuclides that are used in nuclear medicine for SPECT and PET-CT imaging, such as ^89^Zr, ^111^In, ^67/68^Ga, ^64^Cu, and ^52^Mn, and hence has high potential for other therapeutic radionuclides. This property confers the possibility of reliably tracing PLAD biodistribution noninvasively [21] and perhaps better selecting patients for therapy with PLAD. Nuclear medicine studies with radiolabeled liposomes using modern imaging techniques such as PET-CT or SPECT-CT can be extremely helpful in determining the dose distribution to tumors and to predict response in the individual patient. One distinct advantage of nanomedicine is the possibility of co-encapsulating an active pharmaceutical ingredient (API) with an additional agent that can serve as a radionuclide metal chelator to track the nanoparticle and thereby the API biodistribution, as reviewed by Man et al. [47]. PLAD complies with these requirements since Dox is the main API and Ald is a strong chelator of metals such as ^111^In or ^89^Zr [21]. In fact, liposome-encapsulated Ald can serve two purposes: as an immunomodulating agent working in synergy with the co-encapsulated cytotoxic agent doxorubicin and as a carrier of radionuclides useful for imaging liposome biodistribution. This is, in essence, a very relevant example of nanomedicine harnessed for improved theranostics.

The enhanced tumor deposition and retention of drugs delivered by long-circulating liposomes has been well established and is referred to as the enhanced permeability and retention (EPR) effect [48,49,50]. Based on the total tumor measurement of the liposomal drug, it appears that PLAD has an equal or better EPR tumor-targeting effect than PLD. Following liposome extravasation and accumulation in tumor tissue extracellular fluid, the fate of liposomes depends on liposome uptake by the various cell types comprising the tumor parenchyma. It has been recognized that nontargeted liposomes are primarily taken up by TAM or remain in the tumor interstitial fluid in the perivascular zone. Tumor cell uptake of liposomes is relatively low, although it varies depending on the tumor type. Fortunately, in the case of doxorubicin liposomes, liposomes will gradually lose the gradient and release the encapsulated drug in the tumor extracellular fluid, which will thereafter diffuse into surrounding tumor cells and damage them. In this regard, we have recently reported in dissociated tumors that the tumor cell-associated liposomal drug is significantly greater for PLAD than for PLD [18]. This interesting finding may be explained by a faster drug release from PLAD in tumors or by PLAD-induced suppression of TAM activity [51], which may allow for more liposomes to be available for tumor cell uptake.

It is increasingly recognized that therapy aimed at killing cancer cells (i.e., cytotoxic chemotherapy) is insufficient for inducing durable cancer remissions and that mobilization of the adaptive immune response against cancer cells is necessary. Immunotherapies such as immune checkpoint inhibitors can produce complete remission in metastatic cancer patients who remain relapse-free for years [52,53]. Nonetheless, immune checkpoint blockade as a single treatment modality is only efficacious in a small subset of patients, often due to the low tumor immunogenicity and TAM-induced immunosuppression in the tumor microenvironment of most tumors [54,55]. The combination of an immune checkpoint inhibitor with one or more cytotoxic drugs appears to be the most efficacious approach [56,57]. Presumably, as cancer cells are killed by the cytotoxic chemotherapy, they trigger the activation of antigen-presenting cells that synergize with immune checkpoint inhibitors to produce a robust antitumor adaptive immune response. However, combination chemo-immunotherapy is associated with significantly more toxicities, and many cancer patients are unfit and unable to tolerate the addition of chemotherapy due to poor performance status and comorbidities [58]. In this respect, we believe that PLAD has a major potential in chemo-immunotherapy applications. Doxorubicin is a strong immunogenic cell death inducer and, as such, has intrinsic abscopal effects, suggesting it can synergize with immune checkpoint blockade [59]. Furthermore, its encapsulation in liposomes has been shown to significantly reduce drug toxicities in cancer patients. Recently, we showed that alendronate, when encapsulated in liposomes of similar composition to that of PLD, polarized macrophages towards an antitumoral M1-like phenotype and preferentially accumulated in tumor-draining lymph nodes and spleen [60], which are the primary sites for naïve T-cell priming and activation against tumor-associated antigens by antigen-presenting cells such as macrophages. Our studies with PLAD showed increased uptake in the spleen and tumor compared with PLD [18]. Importantly, we found that PLAD shifted cellular drug uptake to TAM and to monocytic myeloid-derived suppressor cells (MDSC) and induced significant changes in the number and functionality of tumor-infiltrating cells, including TAM, MDSC, Treg, natural killer (NK), and NK-T cells [18], which are consistent with enhanced antitumor immune responses in the tumor microenvironment. We believe that the potent tumoricidal and immune-stimulatory effects of PLAD make it superior to PLD or conventional doxorubicin in chemo-immunotherapy regimens.

## 5. Conclusions

Co-encapsulation of ALD and DOX in pegylated liposomes leads to a chemo-immunotherapeutic, multi-modality platform with non-overlapping toxicity and with a unique mechanism of activity that may have a profound impact on cancer therapy. These results open the way for further development of PLAD towards clinical applications of a unique product that blends chemotherapeutic and immunomodulating properties.

## Figures and Tables

**Figure 1 pharmaceutics-15-02606-f001:**
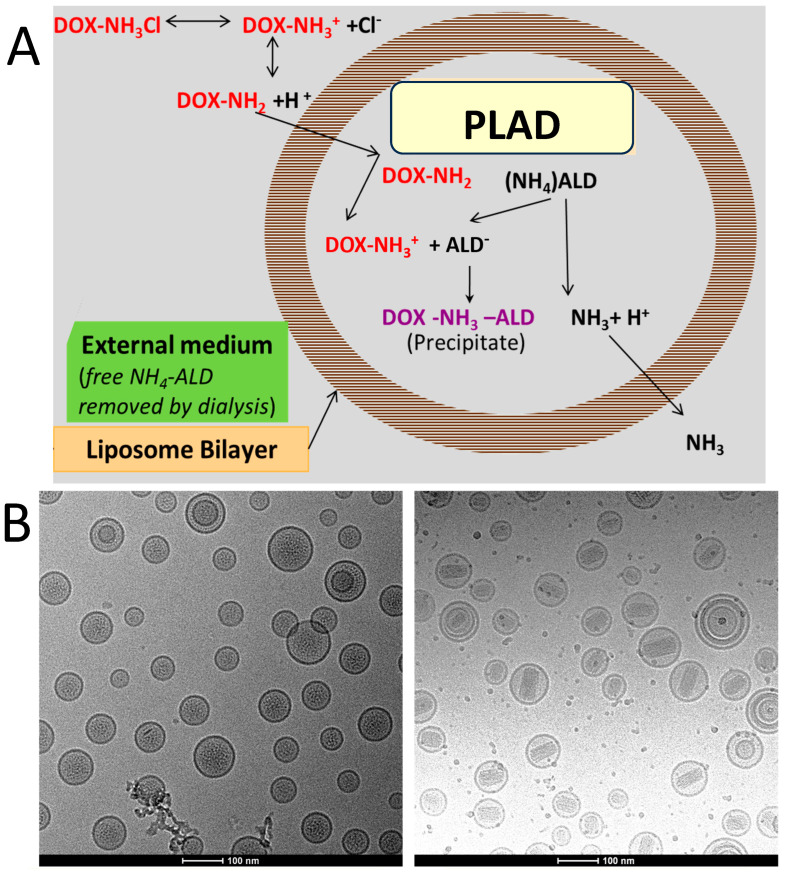
(**A**) Schematic drawing of doxorubicin loading using an ammonium alendronate gradient for co-encapsulation to form Pegylated Liposomal Alendronate salt of Doxorubicin (PLAD). (**B**) Comparative cryoTEM view of alendronate-containing liposomes before (PLA) and after loading with Dox (PLAD). Left panel (PLA): spherical liposomes with few MLV; Right panel (PLAD): spherical liposomes with thick rods of precipitated Dox, and few MLV.

**Figure 2 pharmaceutics-15-02606-f002:**
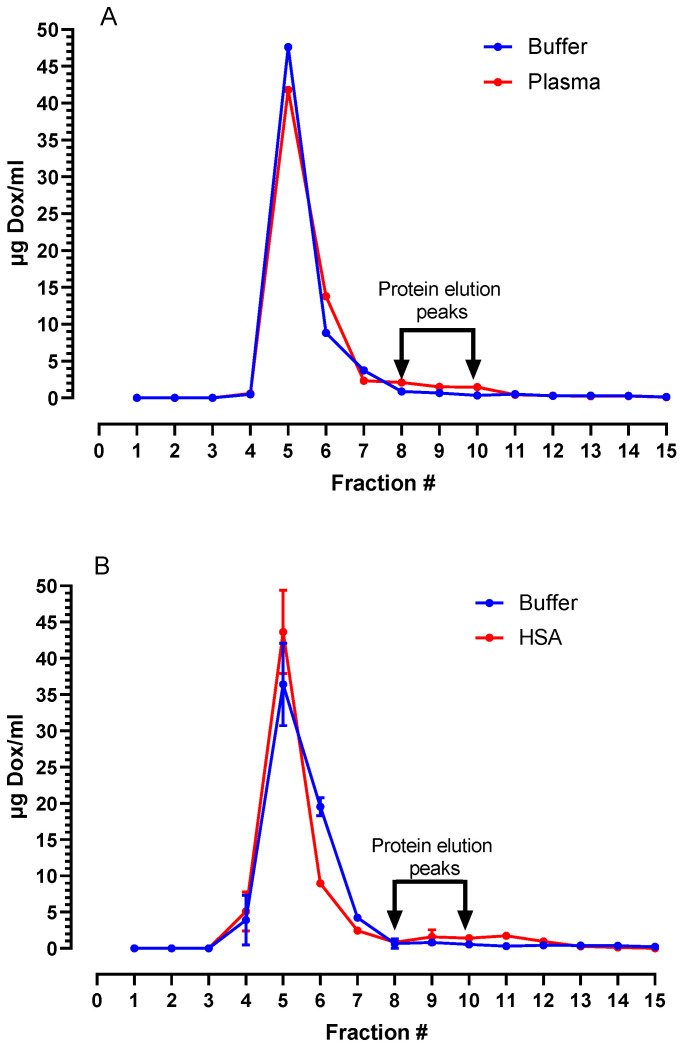
(**A**) Stability of PLAD incubated in 80% human plasma; (**B**) Stability of PLAD incubated in HSA at a concentration of 4 g%. Liposomal drug peak elutes in fractions 5–6, proteins in fractions 8–10, and free drug in fractions 11–12, indicating that Dox remains associated with the liposome fraction, and no significant leakage was detected.

**Figure 3 pharmaceutics-15-02606-f003:**
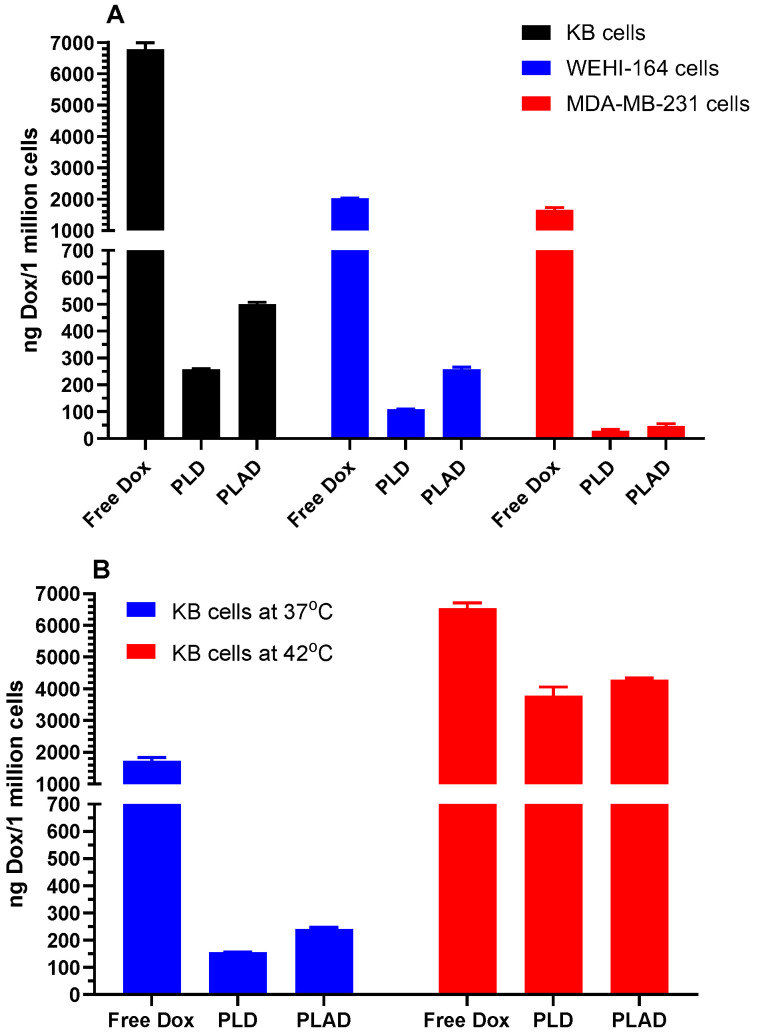
(**A**) In vitro drug uptake by tumor cells exposed to free Dox, PLD, or PLAD at 37 °C for 3 h. As expected, the free drug is taken at much higher levels than liposomal drug. The uptake of liposomal drug per 10^6^ cells varies widely between the different cell lines, with a slightly greater uptake for PLAD than for PLD in KB and Wehi-164 cell lines. (**B**) Effect of temperature increase to 42 °C on drug uptake by KB tumor cells exposed to free Dox, PLD, or PLAD for 3 h. Drug uptake increased with temperature by 3.8-fold for free Dox, 24.2-fold for PLD, and 17.8-fold for PLAD.

**Figure 4 pharmaceutics-15-02606-f004:**
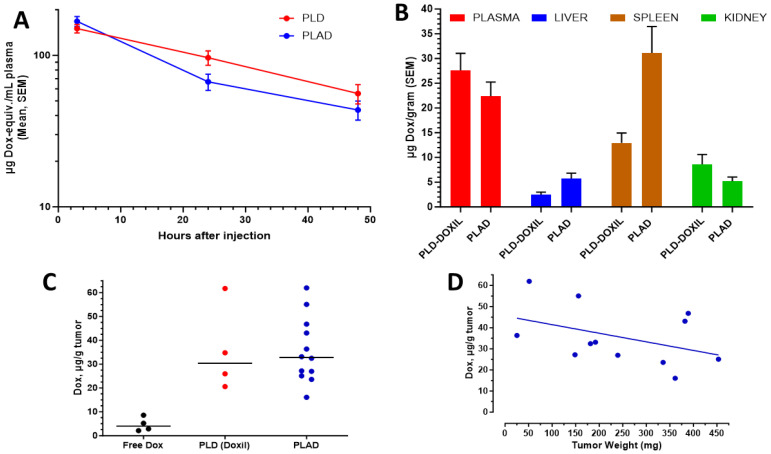
Pharmacokinetics and biodistribution of PLAD and PLD in M109 tumor-bearing BALB/c f mice after i.v. injection at a dose of 10 mg/kg. (**A**) Plasma Dox levels are slightly lower for PLAD than for PLD with a long circulation half-life of ~24 h in both cases. (**B**) Tissue distribution at 72 h post-injection reveals greater liver and spleen drug levels and slightly lower blood levels for PLAD as compared with PLD. (**C**) Tumor drug levels are roughly equivalent for PLD and PLAD and much greater (~10-fold) than in free Dox-injected mice. (**D**) Linear regression plot showing a non-significant trend of lower tumor drug uptake per gram when tumor weight increases.

**Figure 5 pharmaceutics-15-02606-f005:**
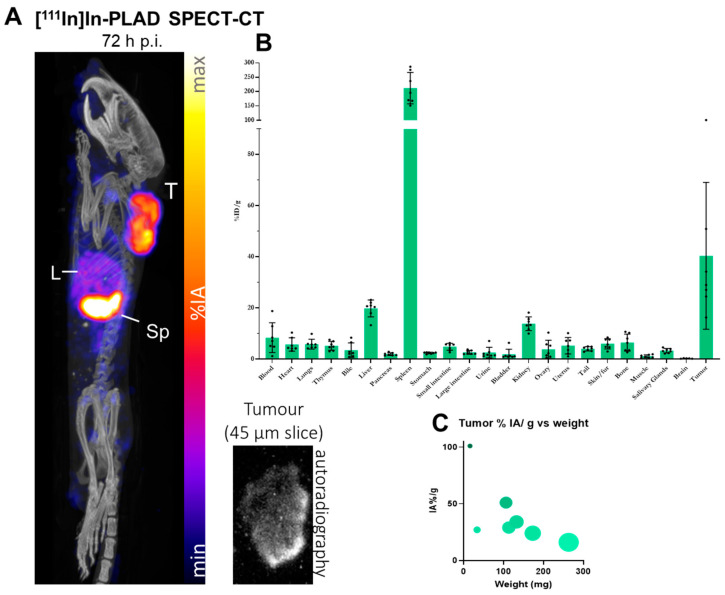
(**A**) Representative images showing SPECT/CT imaging (maximum intensity projection) and autoradiography of a 45 μm tumor slice after 72 h post iv injection of [^111^In]In-PLAD. (T = tumor; L = liver; Sp = spleen; H = heart/blood pool); (**B**) Ex vivo biodistribution of [^111^In]In-PLAD at 72 h post-injection; (**C**) A comparison between tumor uptake of [^111^In]In-PLAD in all tumors vs. their respective mass at 72 h post-injection. The dot size and color intensity represent the tumor size/weight, and the percent tumor uptake (%IA/g) respectively.

**Figure 6 pharmaceutics-15-02606-f006:**
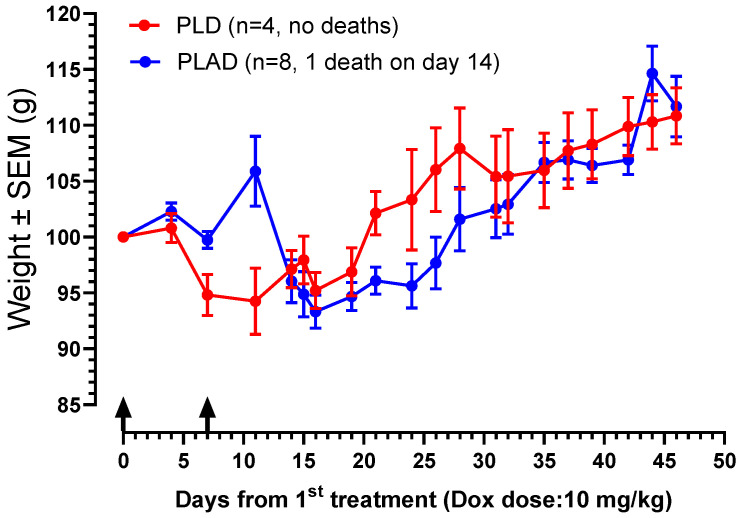
Comparative toxicity of PLD and PLAD in tumor-free Sabra f mice by weight curves. PLD and PLAD injected i.v. at a dose of 10 mg/kg in two successive weekly injections represented by arrows. Mice were weighed at least 2× per week and inspected 3× per week.

**Figure 7 pharmaceutics-15-02606-f007:**
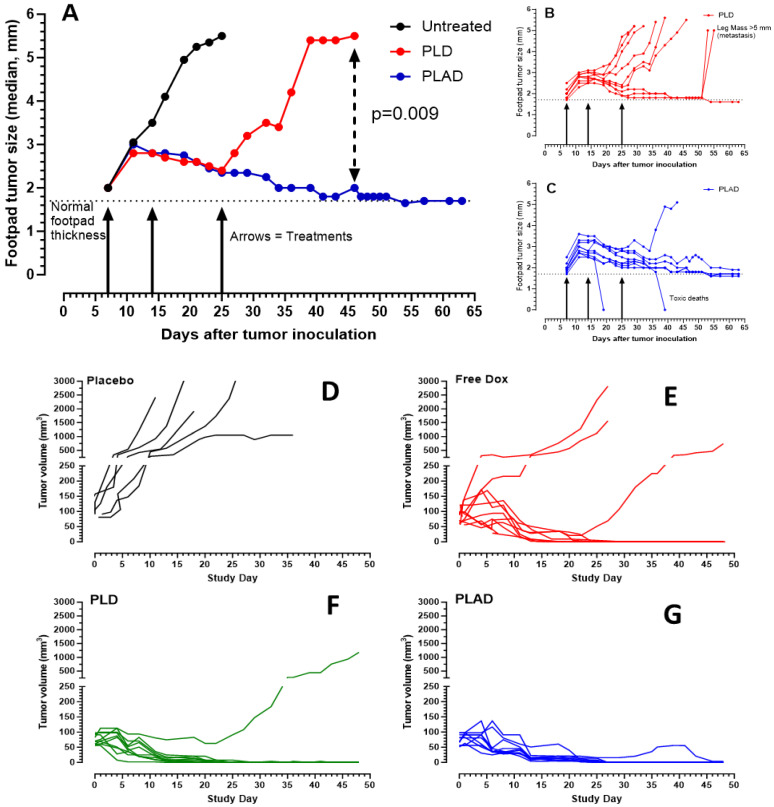
Therapeutic activity of PLAD in 4T1 and WEHI-164 mouse tumor models. (**A**–**C**) 4T1 model: BALB/c inoculated with 10^5^ 471 tumor cells s.c. (intra-footpad). PLD and PLAD injected i.v. at a dose of 8 mg Dox/kg on days 7, 14, and 25. Individual tumor growth curves for PLD and PLAD are presented in panels (**B**,**C**). There were 2 toxic deaths in the PLAD group. At the end of the study, 5/9 mice in the PLAD group were free of tumors, compared with 1/9 in the PLD group. (**D**–**G**) WEHI-164 model: BALB/c mice inoculated with 10^6^ Wehi-164 cells s.c. when tumors reached an estimated volume of 50–100 mm^3^, mice were treated with Placebo (PBS), Free Dox, PLD, or PLAD at a dose of 6 mg/kg weekly x3. Panels (**D**–**G**) present the individual tumor growth curves for each of the treatment groups. All treatments were very effective, although PLAD was the only treatment that achieved complete regression in 100% of the mice. The difference between PLAD and Free Dox curves by the log rank test was borderline significant (*p* = 0.0671).

**Figure 8 pharmaceutics-15-02606-f008:**
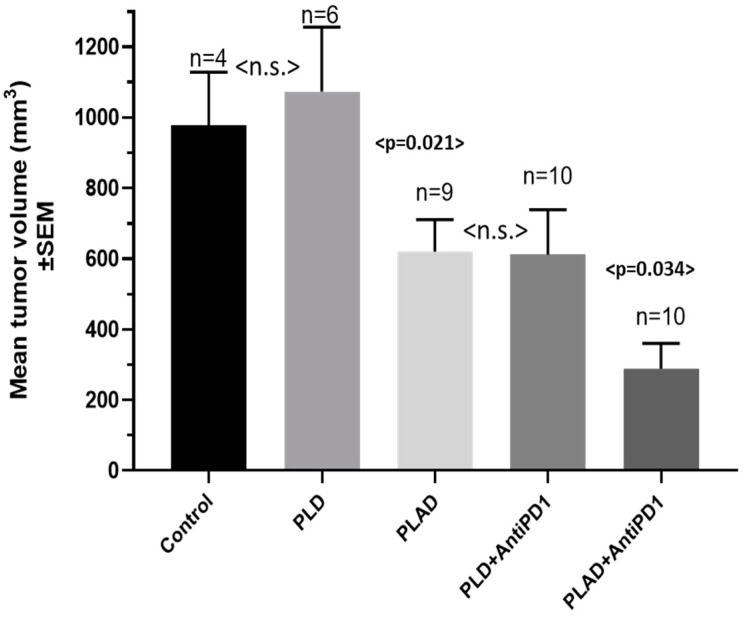
Anti-tumor activity of PLAD and PLD with or without immune checkpoint inhibitors (anti-PD1) in a mouse MDR tumor model (M109R). BALB/c f mice were inoculated s.c. with 10^6^ M109R tumor cells. Once tumors became palpable, mice were treated i.v. with 2 weekly injections of PLD or PLAD at a dose of 8 mg/kg, with or without anti-PD1 at a flat dose of 200 µg per mouse by i.p. injection. On day 40, mice were sacrificed, and tumors were dissected and weighed. The mean tumor weight per treatment group was calculated. Statistical analysis (Mann-Whitney Test): PLD + antiPD1 vs. PLD, *p* = 0.0460; PLAD vs. Untreated, *p* = 0.0539; PLAD vs. PLD, *p* = 0.0211; PLAD + antiPD1 vs. Untreated, *p* = 0.0040; PLAD + antiPD1 vs. PLD, *p* = 0.0069; PLAD + antiPD1 vs. PLD + antiPD1, *p* = 0.0340; PLAD + antiPD1 vs. PLAD, *p* = 0.0144. All other comparisons were not significant. n.s.: not significant.

**Figure 9 pharmaceutics-15-02606-f009:**
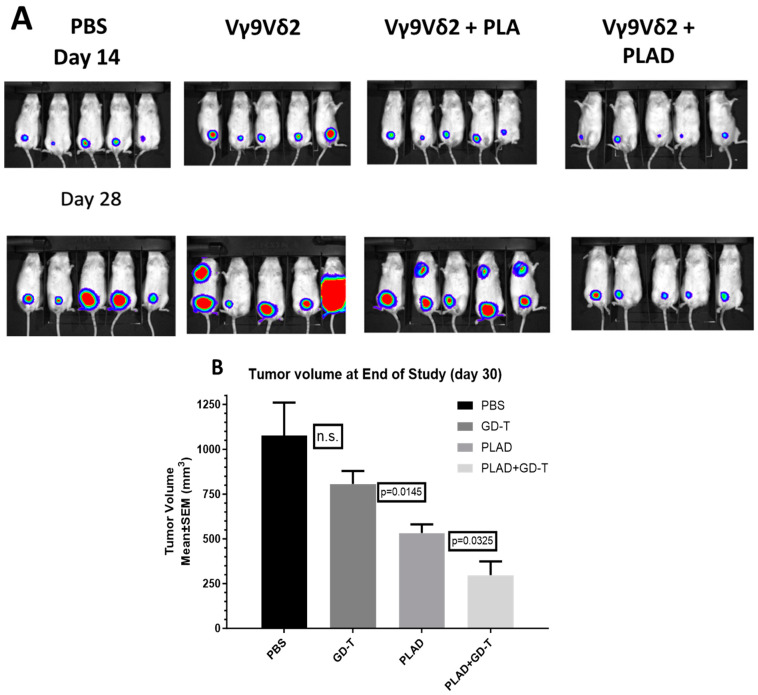
Anti-tumor effect of PLAD in combination with antitumor gamma delta T cells in a triple-negative human breast cancer mouse model. (**A**) SCID Beige mice were inoculated with 10^5^ MDA-MB-231(luc) tumor cells into the lower mammary fat pad. When tumors became palpable, groups of 5 mice each were injected i.v. with PBS control, no treatment, low-dose PLA (5 mg Ald/kg), high-dose PLA (8 mg Ald/kg), or PLAD (5 mg Dox/kg). 72 h later, all mice, except for the PBS group, received by i.v. injection 8 × 10^6^ gamma-delta T-cells per mouse. Bioluminescence imaging (BLI) was followed weekly after i.p. injection of 200 µL luciferin, as previously done [19]. Lymph node metastases can be noticed in some mice injected with PLA and gamma-delta cells. (**B**). A second experiment was done with the same tumor model to compare the efficacy of PLAD with or without gamma delta T cells. In this study, 2 × 10^7^ gamma-delta T-cells were injected 24 h after PLAD 5 mg Dox/kg. Mice were culled on day 30 after treatment started, and tumors were dissected and weighed. PLAD with gamma delta T cells was clearly the most effective treatment. See *p* values (*t* test) in (**B**). n.s.: not significant.

**Table 1 pharmaceutics-15-02606-t001:** Characteristics of PLAD batches used in this study.

PLAD Batch (Batch Size)	Vesicle Size nm	PDI	Zeta Potential mV	Osmolality mOsm/kg	pH	ALD mg/g	Cholesterol mg/g	mPEG2000-DSPE mg/g	HSPC mg/g	DOX-HCl mg/g ^1^
Batch 1 (0.5 L)	110.3	0.058	−12.13	317	7.1	0.5	1.62	1.36	4.6	0.9
Batch 2 (1.5 L)	99.8	0.028	−13.41	291	6.7	0.6	1.62	1.21	4.3	0.9

^1^ The potency of the batch is labeled as 1 mg/mL of Dox-HCl equivalents (acceptable range 0.9–1.1). The actual result in these 2 batches is 0.9 mg/g or ml.

**Table 2 pharmaceutics-15-02606-t002:** In vitro cytotoxicity studies with Free Dox, PLD and PLAD in human and mouse tumor cell lines ^1^.

Cell Line	KB	MDA-MB-231	Wehi-164	4T1	M109
Free Dox	0.07	0.6	0.5	2.2	0.1
PLD	4.8	15.9	18.75	>50	8.0
PLAD-1	0.6	5.1	4.6		1.0
PLAD-2	2.0	11.7	6.6	24.6	
Free Ald			>50		

^1^ Representative results from n = 1–4 experiments per cell line.

## Data Availability

Data supporting reported results is available through the corresponding author.

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
