# Peer review of "Harnessing Nanomedicine to Potentiate the Chemo-Immunotherapeutic Effects of Doxorubicin and Alendronate Co-Encapsulated in Pegylated Liposomes"

_pharmaceutics, 2023, doi:10.3390/pharmaceutics15112606_

Round 1

Reviewer 1 Report

Comments and Suggestions for Authors

The research work “Harnessing nanomedicine to potentiate the chemo-immunotherapeutic effects of doxorubicin and alendronate co-encapsulated in pegylated liposomes” directing to a finding of formulation of Dox and Ald 23 co-encapsulated in pegylated liposomes on large scale production. The work is novel and perfectly in the scope of the journal. The methodology, characterization and results are explained efficiently with necessary data and illustration. The formulations were evaluated for In vitro uptake and cytotoxicity assays and then the radiolabelled formulation was studied for Pharmacokinetics distribution. The illustrations were placed in proper places, and it justifies its necessity.

The comments are:

1.       Line 57-59: The choice of Ald is based on the multifaceted properties of aminobisphosphonates including direct (reduced tumor cell invasion and proliferation) and indirect (reduced osteoclastic activity and angiogenesis) antitumor effects.  Is there any same combination marketed product available? Is it safe to use this combination?

2.       Figure 1B, right side image, clarity/light background should be improved if another image is available.

3.       Line 217: All values fall within the pre-specified acceptable range. How can this be stated?

4.       Line 236-237: The stability of the PLAD formulations was assessed in vitro with a plasma stability assay by exposure to human plasma for 2 h at 37oC. Why was this temperature selected?

5.       Line 244: Maybe sentences break at that point.

Author Response

We thank the reviewer for his comments.

1. As far as we know, there is no other combination in the clinic similar to the Ald-Dox used in our PLAD formulation. Regarding safety, at this point we have only information from rodent studies which seem to indicate that it is slightly more toxic than the Doxil (PLD) commercial formulation when given at near MTD dose. However, at conventional doses, there is no significant difference. 

2. Unfortunately, this is the original image, and we cannot make any changes.

3. The statement above Table 1 ("All values fall within the pre-specified acceptable range") is confusing and has been removed. Instead, we have added an explanatory remark in lines 205-206. 

4. The 37oC temperature was chosen to mimic the in vivo physiologic conditions. This is not related to the shelf stability of the product but to its ability to circulate in stable form in the blood stream.

5. We have reworded the text in lines 243-247 to make it more fluid and understandable

Reviewer 2 Report

Comments and Suggestions for Authors

In this work, the authors used pegylated liposomes (PLAD) to co-encapsulate DOX and ALD, two cancer therapeutic drugs with different therapeutic pathways. PLAD was prepared by rational preparation methods and characterized for its physicochemical properties and stability. The results showed that PLAD had good plasma stability and could be stored for a long time. Subsequently, the anti-tumour efficacy of PLAD and γ-δT cells alone or in combination was evaluated in vitro and in vivo, and the results indicated that PLAD had better anti-tumour efficacy than free DOX and clinical formulation PLD. The above results indicated the potential application of PLAD in cancer chemoimmunotherapy and co-encapsulated nanomedicine. I support the acceptance of this manuscript after the author address the following issues.

1.       For abbreviations, the full name should be given for the first time (e.g., PLAD, TAM).

2.       In the SPECT/CT imaging study method, please indicate the injection administration method.

3.       Some contents in the Formulation and characterization of PLAD section (such as standard preparation methods) should be placed in Materials and Methods.

4.       The last sentence of the second paragraph of the Formulation and characterization of PLAD refers to 'Based on these assays the stability of PLAD liposomes was found to be excellent over the course of ~2 years'. Please provide experimental data and result graphs.

5.       In Figure 1A, remove the grey ellipse on the left and reposition the covered arrow.

6.       In the legend of Figure 2B, please notice the sentence 'Stability of PLAD reduced in HSA a a concentration of 4 g%'.

7.       In Figure 3, Fig3A and Fig3B are represented only by A and B and placed on the left side of the figure.

8.       In Figure 4, please remove the notes above Figure 4C and Figure 4D.

9.       Please unify the format of the journal name abbreviations for all references and use full page numbers (e.g., page numbers in reference 2 should be changed to 1317-1332).

Comments on the Quality of English Language

no comments

Author Response

We thank the reviewer for his/her comments.

  1. We have checked for abbreviations and corrected as requested throughout the manuscript.
  2. We have indicated the method of injection in line 143 (tail vein bolus injection).
  3. We have moved the section on the formulation process to the Methods section as requested (see lines 90-110).
  4. We have reworded this statement to indicate precisely what was tested an to remove a full claim of stability: "Based on this assay and on DLS particle size analysis, no significant leakage of Dox or Ald and no significant change in vesicle size and polydispersity were observed suggesting that these critical parameters of the PLAD formulation are stable over the course of ~2 years (data not shown)."
  5. Figure 1A corrected as requested
  6. Corrected
  7. Corrected
  8. Corrected
  9. We have changed the reference style to a style that complies with the instructions of Pharmaceutics.

Reviewer 3 Report

Comments and Suggestions for Authors

The manuscript” Harnessing nanomedicine to potentiate the chemo-immunotherapeutic effects of doxorubicin and alendronate co-encapsulated in pegylated liposomes” by Gabizon et al. described the bioactive of the dox and ald co-encapsulated in pegylated liposomes (PLAD). T The study provides an important contribution and I recommend that it be accepted for publication of Pharmaceutics.

Author Response

We thank the reviewer for his/her positive comments. There were no revisions requested.

Reviewer 4 Report

Comments and Suggestions for Authors

High-quality work deals with the preparation of liposomal forms of antitumor drugs and evaluation of the efficacy of the obtained systems. Increasing the solubility and bioavailability of doxorubicin is a significant practical task in the field of cancer therapy.

There are no significant comments on the text and format of the paper.

The authors could have added some references to the work on the preparation of coupled liposomes in Pharmaceutics in the text of the introduction. For example,

https://doi.org/10.3390/pharmaceutics14102183

 https://doi.org/10.3390/pharmaceutics15010086

https://doi.org/10.3390/pharmaceutics15020369

 Also, please on pg. 6 to make the font format uniform.

Author Response

We thank the reviewer for his comments.

We have added two of his suggested references in the Introduction section: 

https://doi.org/10.3390/pharmaceutics14102183; https://doi.org/10.3390/pharmaceutics15020369

We have corrected the font size in page 6.